# Searching for SARS-CoV-2 in Cancer Tissues: Results of an Extensive Methodologic Approach based on ACE2 and Furin Expression

**DOI:** 10.3390/cancers14112582

**Published:** 2022-05-24

**Authors:** Sara Ricardo, Pedro Canão, Diana Martins, Ana C. Magalhães, Marina Pereira, Ulysses Ribeiro-Junior, Evandro Sobroza de Mello, Venâncio A. Alves, Regina Pinto, Dina Leitão, Georgina Alves, Rute Oliveira, Joana Wilton, Susete Costelha, Diana Meireles, Didier Cabanes, Leonor David, Fernando Schmitt

**Affiliations:** 1i3S-Instituto de Investigação e Inovação em Saúde, Universidade do Porto, 4200-135 Porto, Portugal; sricardo@ipatimup.pt (S.R.); acmagalhaes@ipatimup.pt (A.C.M.); ldavid@ipatimup.pt (L.D.); 2TOXRUN–Toxicology Research Unit, University Institute of Health Sciences, CESPU, CRL, 4585-116 Gandra, Portugal; 3Medical Faculty, University of Porto, 4200-319 Porto, Portugal; dinaraquel@med.up.pt; 4Department of Pathology, Centro Hospitalar e Universitário de São João, 4200-319 Porto, Portugal; pedro.a.canao@gmail.com (P.C.); dianafnfm@gmail.com (D.M.); 5ICBAS–Instituto de Ciências Biomédicas Abel Salazar, Universidade do Porto, 4050-313 Porto, Portugal; 6Instituto do Câncer, Hospital de Clínicas (HCFMUSP), Faculdade de Medicina, Universidade de São Paulo, São Paulo 01246000, Brazil; ma.ap@hotmail.com (M.P.); ulysses.ribeiro@fm.usp.br (U.R.-J.); esobroza@gmail.com (E.S.d.M.); venancio@uol.com.br (V.A.A.); 7Ipatimup–Institute of Molecular Pathology and Immunology, University of Porto, 4200-465 Porto, Portugal; rpinto@ipatimup.pt; 8i3S Diagnostics–Instituto de Investigação e Inovação em Saúde, Universidade do Porto, 4200-135 Porto, Portugal; georgina.alves@ibmc.up.pt (G.A.); rute.oliveira@ibmc.up.pt (R.O.); joana.wilton@i3s.up.pt (J.W.); susete_costelha@ibmc.up.pt (S.C.); diana.meireles@ibmc.up.pt (D.M.); didier@ibmc.up.pt (D.C.); 9CINTESIS@RISE, Alameda Professor Hernâni Monteiro, 4200-319 Porto, Portugal

**Keywords:** SARS-CoV-2, ACE2, cancer tissues, infection tropism

## Abstract

**Simple Summary:**

SARS-CoV-2 is the virus that causes COVD-19; there is consensus that this virus infects cells presenting the angiotensin-converting enzyme 2 (ACE2) receptor at the cell surface. In our study, we assessed the expression profiles of cancer cells regarding the expressions of ACE2 and furin (another important player in the infection process). We analyzed a series of formalin-fixed paraffin-embedded samples of tumor tissues from cancer patients who underwent surgery at the beginning of the pandemic, including some cases with known positive results for SARS-CoV-2. Our goal was to explore the possibility of viral infections in cancer tissues by detecting viral RNA and/or proteins using histology-based methods. From this extensive methodologic approach, we show that colon and gastric carcinoma cells present high expression levels of ACE2 and furin, contrasting with a negative expression profile of ACE2 in thyroid carcinoma. Some cases tested positive for SARS-CoV-2 by PCR extracted from tissue sections, but in situ hybridization and immunohistochemistry did not show consistent results of a viral presence in the cancer cells. Our study raises the possibility of ACE2-mediated viral tropism for cancer tissues to be clarified in future studies.

**Abstract:**

SARS-CoV-2 pandemics have been massively characterized on a global scale by the rapid generation of in-depth genomic information. The main entry gate of SARS-CoV-2 in human cells is the angiotensin-converting enzyme 2 (ACE2) receptor. The expression of this protein has been reported in several human tissues, suggesting a correlation between SARS-CoV-2 organotropism and ACE2 distribution. In this study, we selected (a series of) 90 patients who were submitted to surgery for tumor removal between the beginning of the SARS-CoV-2 pandemic and the closure of operating rooms (by the end of March 2020) in two different countries—Portugal and Brazil. We evaluated the expressions of ACE2 and furin (another important factor for virus internalization) in colon (*n* = 60), gastric (*n* = 19), and thyroid (*n* = 11) carcinomas. In a subseries of cases with PCR results for SARS-CoV-2 detection in the peri-operatory window (*n* = 18), we performed different methodological approaches for viral detections in patient tumor samples. Our results show that colon and gastric carcinomas display favorable microenvironments to SARS-CoV-2 tropism, presenting high expression levels of ACE2 and furin. From the subseries of 18 cases, 11 tested positive via PCR detection performed in tumor blocks; however, a direct association between the ACE2 expression and SARS-CoV-2 infection was not demonstrated in cancer cells using histology-based techniques, such as immunohistochemistry or in situ hybridization. This study raises the possibility of ACE2-mediated viral tropism in cancer tissues to be clarified in future studies.

## 1. Introduction

The spread of severe acute respiratory syndrome coronavirus 2 (SARS-CoV-2) has taken the world by surprise. COVID-19, an infectious disease caused by SARS-CoV-2, is transmitted by a coronavirus similar to Severe Acute Respiratory Syndrome (SARS) and the Middle East respiratory syndrome (MERS). To date, COVID-19 has caused over 5 million deaths worldwide [1,2]. Due to the emergence of this infection, researchers are striving to understand the mechanisms of the infection, the best treatment options, and vaccine production.

Coronaviruses (CoVs) belong to the family Coronaviridae, which are enveloped and positive-strand RNA viruses [3]. The SARS-CoV-2 genome sequence shares an ~80% sequence identity with SARS-CoV and ~50% with MERS-CoV [4,5]. Its genome comprises 14 open reading frames (ORFs), two-thirds of which encode 16 non-structural proteins that make up the replicase complex [5,6]. The remaining one-third encodes nine accessory proteins (ORF) and four structural proteins: spike (S), envelope (E), membrane (M), and nucleocapsid (N), of which Spike mediates SARS-CoV entry into host cells [7]. The S gene of SARS-CoV-2 is highly variable from SARS-CoV, sharing <75% nucleotide identity [4,5,8]. Spike has a receptor-binding domain (RBD) that binds strongly to angiotensin-converting enzyme 2 (ACE2) and an S1/S2 polybasic cleavage site that is proteolytically cleaved by cellular cathepsin L and the transmembrane protease serine 2 (TMPRSS2) [8,9]. It was also reported that the Spike glycoprotein of SARS-CoV-2 harbors, at the boundary between its two subunits (S1/S2), a furin cleavage site that is activated by the host cell enzyme furin proprotein convertase to increase the virus affinity to its primary receptor ACE2 [10,11]. Furin also helps the virion to efficiently bind with ACE2 via the receptor-binding domain (RBD) S1 of the S-protein, to transmit the virus as a stable form and to invade the host cell rapidly for further pathogenicity [11,12,13]. Proteolytically-active furin is abundantly expressed in human bronchial epithelial cells and constitutes the basis for cellular tropism of SARS-CoV-2 infection [13].

Although the main targets of SARS-CoV-2 are cells from the respiratory tract, which constitutively express ACE2, multi-organ dysfunction has also been reported in patients with COVID-19. The SARS-CoV-2 spike protein was previously detected in the kidney, small intestine, pancreas, blood vessels, and other organs by immunofluorescence in a patient who died from COVID-19. In addition, the SARS-CoV-2 spike protein has also been found in co-localization with ACE2, suggesting a correlation between SARS-CoV-2 organotropism and ACE2 distribution [14].

Accordingly, our study investigates whether SARS-CoV-2 could be present in cancer tissue expressing ACE2, based on previous data showing that tumor cells can express this receptor for the virus [14]. The specific goals were to evaluate the expression patterns of ACE2 and furin in cancer cells and normal-adjacent tissues in colorectal, gastric, and thyroid carcinomas; to investigate if SARS-CoV-2 is able to infect cancer cells, and to see if this is associated with the expressions of ACE2 and furin.

We selected (a series of) patients who were submitted to surgery for tumor removal between the beginning of the SARS-CoV-2 pandemic and the closure of operating rooms (by the end of March 2020). Retrospectively, we obtained information from SARS-CoV-2 tests performed in the peri-operatory period. To investigate the presence of SARS-CoV-2 in tumor samples, we tested different methodological approaches for viral detection (Immunohistochemistry (IHC), in situ hybridization (ISH), and RT-qPCR) in formalin-fixed paraffin-embedded tumor samples.

## 2. Materials and Methods

### 2.1. Patient Selection

This study included patients with cancer at any stage who underwent surgery at Centro Hospitalar de São João (Porto, Portugal) (*n* = 58) (date of surgery between February and May 2020) and Instituto do Cancer do Estado de Sao Paulo—ICESP-HCFMUSP (Sao Paulo, Brazil) (*n* = 32) (date of surgery between March and September 2020). A total of 90 patients (47 males and 43 females; age range 16–92) were studied. According to the tumor site, 11 (12.2%) corresponded to thyroid carcinomas, 19 (21.1%) gastric cancer, and 60 (66.7%) colon carcinomas. Clinical information was obtained by reviewing the clinical processes and all diagnoses were confirmed histologically. The majority of colon carcinomas were classified as adenocarcinomas NOS, gastric carcinomas were of the intestinal type or mixed tumors, and thyroid carcinomas were all papillary types.

This study was conducted in accordance with the Declaration of Helsinki. The protocol was approved by the Ethics Committee of Hospital das Clínicas da Faculdade de Medicina da Universidade de São Paulo, Brazil (project identification code: 4.338.488). The use of retrospective material from human samples for research purposes in Portugal is legally foreseen in the Portuguese law (Decreto-Lei n. º 274/99).

### 2.2. Cell Line Infection

VERO cells (ATCC, no. CCL-81; ATCC, Manassas, VA, USA) were propagated in Dulbecco’s modified eagle medium (DMEM—Thermo Scientific, Waltham, MA, USA) supplemented with 10% fetal bovine serum (FBS—Labclinics, Barcelona, Spain), 100 U/mL of penicillin, and 100 ug/mL of streptomycin. Cells were maintained at 37 °C under a 5% CO_2_ humidified atmosphere.

VERO cells were seeded for 24 h prior to infection in T-75 flasks at a density of 2 × 106/flask. On the infection day, the cell culture media in each flask was discarded and cell monolayers were inoculated in-house [15]. SARS-CoV-2 was isolated with a multiplicity of infection (MOI) of 0.01. Viral infection was allowed to proceed for 1 h at 37 °C in 5% CO_2_; after that, DMEM with 2% of FBS was added to the flasks.

Control and infected cells were independently collected by scraping the cells from the flask with PBS 1×, followed by centrifugation and fixation with 10% neutral-buffered formalin. After fixation, each cell pellet was embedded in Histogel (Thermo Scientific, Waltham, MA, USA) according to the manufacturer’s instructions, followed by standard histological processing and paraffin embedding.

### 2.3. Electron Microscopy

To uncover the infection pattern of SARS-CoV-2 in cultured cells, a qualitative assessment of cell ultrastructure under electron microscopy analysis was carried out. Briefly, for transmission electron microscopy (TEM) and semithin section analysis, cultured cells were grown until 80% confluence and collected from the flask after 3 washes with PBS. Then, cells in PBS were gently scraped from culture dishes and fixed overnight in a solution of 2.5% glutaraldehyde and 2% paraformaldehyde in 0.1 M sodium cacodylate buffer (1:1), at 4 °C. After fixation, cells were pelleted by centrifugation (1200 rpm, 5 min) and washed 3 times in 0.1 M of sodium cacodylate buffer for 5 min. A post-fixation in 0.1 M sodium cacodylate buffered 2% OsO_4_ was carried out for 2 h at room temperature. Then, cells were washed in distilled water 3 times for 5 min each. Incubation with 1% uranyl acetate for 30 min at room temperature at 4 °C was carried out, followed by 3 washes in distilled water for 5 min each. The pellet was included in Histogel™ (Thermo, Waltham, MA, USA, HG-4000-012). After dehydration in ethanol, the pieces were embedded in an epoxy resin. Ultrathin sections were stained and observed in a JEOL 100CXII transmission electron microscope, operated at 60 kV.

### 2.4. Immunohistochemistry

First, we accessed the expression profiles of cellular mediators of virus infection, ACE2, and furin, by IHC. IHC was performed on the Discovery ULTRA staining instrument (Ventana Medical Systems). Briefly, tissue sections were deparaffinized with EZ Prep solution (Ventana Medical Systems, cat no. 950-102) for 16 min at 72 °C. Afterward, heat-induced epitope retrieval using CC1 solution (Ventana Medical Systems, cat no. 950-124) was run for 64 min at 95–100 °C. Then, endogenous peroxidase activity was blocked using a DISC inhibitor reagent (Ventana Medical Systems, cat no. 760-4840). Sections were covered with a primary antibody at a pre-defined dilution (ACE2 dilution 1:2500 (clone CL4035), Thermo Fisher, Waltham, MA, USA) and Furin dilution 1:50 (clone EPR 14674, Abcam, Cambridge, UK) and incubated at 37 °C for 15 min. We used the Ventana Detection Kits: ultraView Universal 3,3′-diaminobenzidine (Ventana Medical Systems, cat no. 760-500) according to the manufacturer’s instructions. We counterstained all sections with hematoxylin II (Ventana Medical Systems, cat no. 790-2208) for 8 min, followed by Bluing Reagent (Ventana Medical Systems, cat no. 760-2037) for 4 min. Normal kidney tissue and the infected VERO cell line were used as positive controls. Immunocytochemistry results were evaluated by, at least, two independent observers (SR, FS, DM, and/or PC) who registered the staining intensity ((0) no staining; (1) week staining; (2) moderate staining; and (3) strong staining) and the percentage of cells stained ((0) 0%; (1) 0–1%; (2) 2–10%; (3) 11–33%; (4) 34–66%; (5) 67–100%). Then, a score was calculated by summing the intensity and extension values. We considered a score < 2 as negative and a score > 3 as positive.

For SARS-CoV-2 protein detection, we performed IHC using primary antibodies against the spike protein (dilution 1:100) (Clone 1A9, GeneTex) and nucleocapsid protein (dilution 1:100) (Clone 6H3, GeneTex), with the same automated procedure described above. The infected VERO cell line (from monkey kidney) was used as a positive control.

### 2.5. RNA in Situ Hybridization

To detect SARS-CoV-2 genomic RNA in FFPE tissues, ISH was performed using the RNAscope 2.5 HD Brown kit (Advanced Cell Diagnostics) according to the manufacturer’s instructions and previously published data [15]. Tissue sections were deparaffinized with xylene, underwent a series of ethanol washes and peroxidase blocking, and were then heated in a kit-provided antigen retrieval buffer and digested by a kit-provided proteinase. Sections were exposed to ISH target probe pairs (RNAscope^®^ Probe-V-nCoV2019-S) and incubated at 40 °C in a hybridization oven for 2 h. After rinsing, the ISH signal was amplified using a kit-provided pre-amplifier and an amplifier conjugated to peroxidase and incubated with a DAB substrate solution for 10 min at room temperature. Sections were then stained with hematoxylin, air-dried, mounted, and stored at 4 °C until image analysis. A formalin-fixed paraffin-embedded infected VERO cell line was used as the positive control (Appendix A).

### 2.6. RNA Extraction and RT-qPCR

From each selected FFPE sample, two sections (10-µm of thickness) adhered to the glass slides were obtained for RNA extraction; an extra 3-µm section was cut and stained with Hematoxylin–Eosin for an adequate evaluation of the tissue. Between each sample cutting, the microtome used was carefully decontaminated with RNase AWAY™ Surface Decontaminant from Thermo Scientific™, and a new blade was used to prevent cross-contaminations. RNA extraction and purification were performed using Maxwell^®^ RSC RNA FFPE Kit and Maxwell^®^ RSC Instrument, from Promega (Madison, WI, EUA), according to the manufacturer’s instructions; except for the elution volume, samples were eluted in 40 µL of the final volume.

PCR amplification of SARS-CoV-2 genes was performed using the Fosun Novel Coronavirus (2019-nCov) RT-PCR Detection Kit (reference PCSYHF02-a, Fosun), adapted from the manufacturer’s instructions, and using 10 µL of RNA as input. Briefly, for a 30-μL reaction, 14.58 µL of 2019-nCov Reaction Reagent was combined with 6.25 µL RT-PCR Enzyme, RNA sample, and water to a final volume of 30 μL. Human RNaseP primers and probe (RNase P (ATTO™ 647), cat no. 10,006,836 (forward primer), 10,006,837 (reverse primer), and 10,007,061 (probe), IDT) were added to the master mix as the extraction control, respectively, at 400 nM (primers) and 100 nM (probe) of the final concentration. The PCR program was as follows: 50 °C for 15 min, 95 °C for 3 min, and 50 cycles of 95 °C for 5 s and 60 °C for 40 s.

Criteria for SARS-CoV-2 positivity in samples were as follows: if at least two out of the three viral genes—E (ROX channel), N (VIC channel), ORF1ab (FAM channel)—had Ct ≤ 36.5, the sample was considered positive for SARS-CoV-2. If only one gene had Ct ≤ 36.5, the sample was considered inconclusive. If all three viral genes were undetected, or the Ct was >36.5, the sample was considered negative. RNaseP RNA was detected using the Cy5 channel.

Three controls were included in the qPCR assay: a no-template control (nuclease-free H_2_O), a negative control (human WT control DNA, reference 4451855, Ambion, Austin, TX, USA), and a positive control (included in the kit). Good laboratory practices were maintained to limit contamination, such as a dedicated pre-PCR room, regular decontamination of pipettes and work surfaces, and the use of filtered pipette tips and aliquoting reagents to single-use volumes.

### 2.7. Statistical Analysis

A statistical analysis was calculated using SPSS statistics 20.0 software (SPSS Inc., Chicago, IL, USA). Fisher’s exact test was applied to compute the *p*-value of the associations in Table 1 and Appendix A; Pearson’s chi-squared test was applied to assess the association between different clinicopathological features and ACE2 and furin expressions. In the two-tailed testing, a *p*-value < 0.05 was considered statistically significant.

## 3. Results

### 3.1. ACE2 and Furin Are Highly Expressed in Colon and Gastric Carcinomas

By IHC, we assessed the expression of ACE2 and furin in 90 formalin-fixed paraffin-embedded carcinoma samples: colon (*n* = 60), gastric (*n* = 19), and thyroid (*n* = 11). ACE2 expression was frequently observed in colon (52/60, 86.7%) and gastric carcinomas (14/19, 73.7%), but never observed in thyroid carcinomas (0/11) (Figure 1A). Furin expression was observed in all thyroid carcinoma cases (11/11, 100%) and frequently identified in gastric (16/19, 84.2%) and colon carcinomas (52/60, 86.7%) (Figure 1B).

ACE expression was observed in the cytoplasm and cell membrane of tumor cells and furin staining was limited to the cytoplasm in a granular pattern (Figure 2).

Since simultaneous expressions of ACE2 and furin facilitate SARS-CoV-2 entry in the host cells, we also explored the association between both proteins in our series. The simultaneous expressions of furin and ACE2 occurred predominantly in colon and gastric carcinoma samples, corresponding to 81.7% (49/60) and 68.4% (13/19) of the cases, respectively (Table 1). Regarding the clinical and pathological data (i.e., tumor size, lymph node metastasis, distant metastasis, residual tumor, invasion, and presence of inflammatory infiltrate) we did not find a significant difference between the ACE2 and furin expressions and the clinicopathological features of the cases.

Comparing adjacent and tumor tissues in the different tumor locations, the positivity for ACE2 was similar in both contexts (non-neoplastic colon mucosa 50/60 (92.6%), non-neoplastic gastric mucosa 16/19 (84.2%), and normal thyroid follicles 0/11 (0%)) (Appendix A). Moreover, furin expression was observed in non-neoplastic tissues of the colon (41/60, 78.8%), stomach (19/19, 100%), and thyroid (10/19, 90.9%) in percentages that did not significantly differ from cancer tissue. ACE2 was highly expressed in areas of gastric intestinal metaplasia (Appendix A).

### 3.2. Different Methodologic Approaches to Detect SARS-CoV-2 in Tumor Samples

To detect the presence of SARS-CoV-2 in our series of FFPE samples, we used specific monoclonal antibodies to detect different viral proteins (spike protein and nucleocapsid); specific RNA probes do detect viral RNA by in situ hybridization and in extracted RNA for RT-qPCR. We selected 18 cases from patients with information about the infection status at the time of surgery to check for the presence of the virus in cancer tissues. As a control, we used infected VERO (renal monkey cell line) cells and performed fixation and histological processing to obtain a control with the same pre-analytical conditions. This control was very effective for antibody and in situ hybridization condition standardization and allowed the visualization of the viral proteins and viral RNA sequences in infected cells (Figure 3 and Appendix A). Additionally, another control was used, a FFPE placenta from a vertical infection case where we detected high levels of SARS-CoV-2 by IHC and ISH (Appendix A). We also performed ultra-structural studies in infected VERO cells to fully validate our control system, and nicely-identified SARS-CoV-2 viral particles (Appendix A). After this initial validation step, we assessed the presence of viral proteins and viral RNA in tumor samples (Figure 3), always including a positive control of infected VERO cells in each slide.

The subset of 18 cases we selected was 9 positive and 9 negative for SARS-CoV-2, by a nasopharyngeal PCR test in the peri-operatory period. IHC results using anti-spike and anti-nucleocapsid antibodies revealed a concordance between them, with 4/18 cases presenting staining with both antibodies and 11/18 cases negative for both antibodies (Appendix A).

From the patients with positive tests, we observed immunostaining for viral antigens in 4/9 (44.4%) (Figure 4). Additionally, we observed staining with SARS-CoV-2 antibodies in 2 cases with negative PCR tests before surgery (2/9, 22.2%). The staining was focal and more intense in the apical membrane of tumor cells (Figure 3). In situ hybridization was negative in all cases despite the positivity in the VERO cell line controls included in every slide (Figure 3). We also detected unspecific staining in adjacent tissues, such as smooth muscle fibers, vessel walls (endothelium and muscle), and cells from connective tissues (fibroblasts) (Appendix A).

To explore the presence of viral RNA particles in tumor tissue, we further performed quantitative PCR in 18 cases (Figure 4). Of nine cases that tested positive before surgery, eight (88.9%) tested positive in FFPE tumor blocks and one (11.1%) tested negative. On the other hand, from the nine cases that were PCR-negative before surgery, three (33.3%) tested positive in tumor tissues and three (33.3%) were inconclusive.

The 18 tumors analyzed for viral presence in FFPE tissue (Figure 4) were all positive for furin; 15/18 (83.3%) were also positive for ACE2. There were no differences in the expressions of ACE2 and furin between positive or negative cases for viral RNA by PCR.

## 4. Discussion

We observed high levels of expression of the ACE2 receptor in colon and gastric non-neoplastic tissues and a particularly striking high expression in gastric intestinal metaplasia. This observation may explain gastrointestinal (GI) symptoms, such as diarrhea, nausea, and vomiting, which have been reported as COVID-19-related symptoms [16], namely by lending support to the known pathophysiology associated with the presence of ACE2 receptors, identified throughout the GI tract [17]. Our results regarding ACE2 expression in non-neoplastic tissues adjacent to adenocarcinomas showed data similar to those recently described by Hikmet et al. [18], although we observed a curiously highly-increased expression of ACE2 in gastric intestinal metaplasia. A recent study by Zhang et al. [19] had a similar observation using RNA sequencing and suggested that intestinal metaplasia, associated with Helicobacter pylori infection, might facilitate SARS-CoV-2 entry in the gastric mucosa.

By observing thyroid dysfunctions in COVID-19 patients, there are indirect indications that thyroid cells are vulnerable to SARS-CoV-2. Baldelli et al. described low serum T3 levels in infected patients but this hormonal imbalance was ascribed to non-thyroid disease secondary to infection [20]. However, several studies on normal thyroid tissues demonstrated that ACE2 is expressed at the mRNA level [21,22,23]. Narayan et al. examined the ACE2 expression in thyroid tissues by IHC and described staining in the cytoplasm of thyroid cancer cells [24]. Using IHC, we could not confirm ACE2 expression in normal thyroid tissues, and expression in cancer tissues was restricted to 11% of the cases. Results are therefore contradictory and limited by the small numbers of series and samples analyzed, and further complicated by the use of different approaches for viral detection.

Existing evidence from RNA-based analyses demonstrates that several cancer types present high mRNA levels of ACE2 [25] but morphological studies are still lacking. The high expression of ACE2 we observed in colon and gastric cancer is in agreement with the recent identification (using a bioinformatics approach in a TCGA pan-cancer panel) of ACE2 overexpression in colorectal and gastric carcinomas [26], supporting that these two types of cancer tissues could internalize SARS-CoV-2. To gain more insight into this hypothesis, we evaluated the possible coexistence of furin in the cancer tissues likely cooperating with ACE2 for viral entry. We identified a high expression of furin in all cancer types, implying that most cancers expressing ACE2 also had furin expression, therefore creating exceptionally favorable conditions for SARS-CoV-2 infection of cancer cells. A model where the coexistence of ACE2 and furin created a particularly favorable environment for SARS-CoV-2 infection in trophoblast cells of the human placenta has been described [27]. Therefore, we departed to the evaluation of SARS-CoV-2 in cancer tissues with the expectation of positive observations.

Our results with anti-spike and anti-nucleocapsid antibodies in tumor samples were not convincing. Unspecific staining in adjacent tissues was troubling and, further, the IHC positivity only matched with five cases (5/9, 55.6%) of patients with positive PCR tests, and was surprisingly observed in two cases that had tested negative in oropharyngeal swabs at the perioperative period. The production of specific antibodies is essential to identify, with precision, the protein epitope of interest. In fact, IHC detection of SARS-CoV-2 is still under scrutiny and antibody specificity is limited, with immunostaining being reported in negative control tissues [28]. The main pillar for staining specificity in our current study was the use of infected VERO cells, which gave us a gold standard (included in every slide) for the specificity of viral recognition.

Our findings deceivably failed to demonstrate the presence of SARS-CoV-2 RNA in tumor samples by ISH. The methodology has been used successfully in FFPE samples [29]; in our placenta and infected VERO cell controls we obtained very consistent and specific staining. The most plausible interpretation for negative results in all our FFPE carcinoma cases is that ISH is very sensitive to fixation conditions and/or that it only detected a minimum viral load that was not attained in our samples. IHC and ISH allow for the evaluation of SARS-CoV-2 presence in specific tissue and cell localization, an asset which could not be attained by RT-qPCR that is performed in RNA extracted from tumor tissue blocks with tumor microenvironment components and residual adjacent normal tissues. Despite this limitation, we had to rely on this approach for our analysis.

The six cases that tested positive or inconclusive in qPCR from the surgical specimen despite a negative nasopharyngeal test should cautiously be considered as real discrepancies since the nasopharyngeal test was performed two days before surgery and, therefore, an intra-hospital infection cannot be discarded. Moreover, the negative nasopharyngeal test does not exclude the presence of a SARS-CoV-2 infection because the sample collection at the nasopharynx could have been performed during an early stage of infection, translating into low viral loads and, subsequently, undetectable RNA levels. In addition, qPCR tests at the hospital and on tumor samples were performed on different materials whose relative quantities were difficult to estimate using qPCR kits with different efficiencies/sensitivities.

A major caveat of the current study is the selection of cases. We selected patients operated on during the initial months of the pandemic in 2020, but only confirmed peri-operative COVID-19 detection in oropharyngeal swabs in nine cases. However, at the time under scrutiny, it could well be that undetected COVID-19 cases were operated on without triage, so we decided to include all cases of surgical resection during the time frame.

## 5. Conclusions

To address our initial hypothesis of a role in the tumor expression of ACE2 and furin in cancer tissues, we defined a previously unrecognized profile of the expressions of ACE2 and furin in cancer tissues, which is interesting, per se, and may impact tumor biology in relation to SARS-CoV-2 infection. Our results show that colon and gastric carcinomas display a favorable microenvironment to SARS-CoV-2 tropism, presenting high expression levels of ACE2 and furin. In fact, 11 of 18 cases evaluated for viral presence had positive PCR detections of SARS-CoV-2, and 10 of these were positive for ACE2 and furin. The series is too small to allow any significant conclusion, but a window of observation is now open to future studies. Moreover, the high levels of expression of ACE2 in intestinal metaplasia of the gastric mucosa deserve further attention in future studies. Finally, although a direct association between ACE2 expression and SARS-CoV-2 infection was not demonstrated in cancer cells, our study raises the possibility of clarifying an ACE2-mediated viral tropism when more reliable IHC or ISH studies are available.

## Figures and Tables

**Figure 1 cancers-14-02582-f001:**
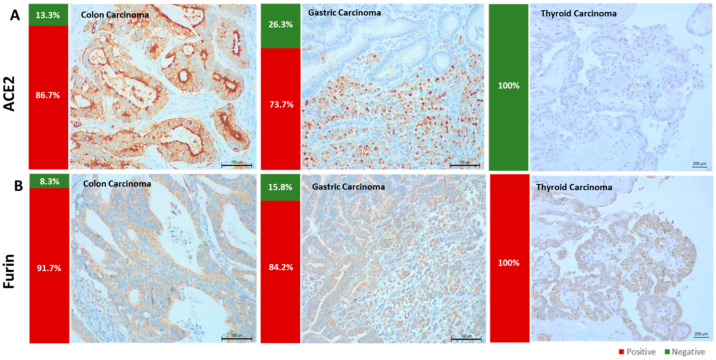
ACE2 and furin expressions in colon, gastric, and thyroid carcinomas. (**A**) ACE2 was highly expressed in colon and gastric carcinomas but not expressed in thyroid tumors. (**B**) Furin was expressed in all carcinoma types evaluated.

**Figure 2 cancers-14-02582-f002:**
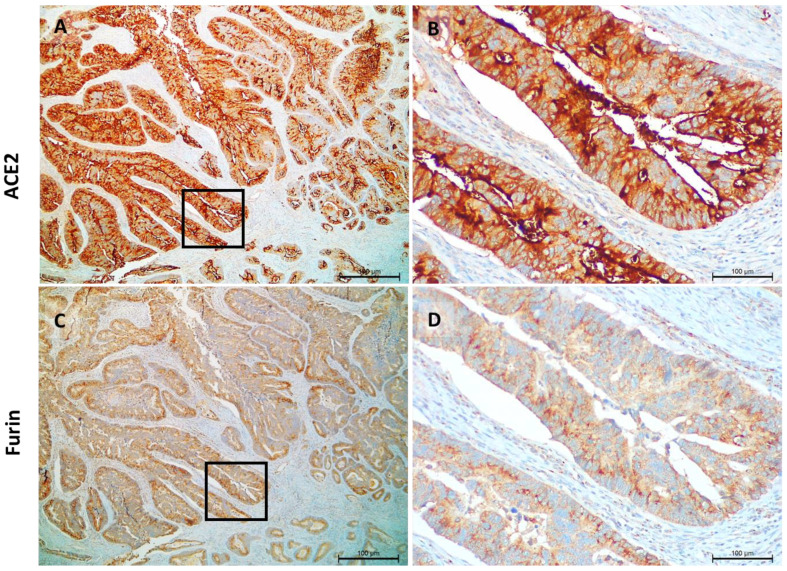
Pattern of expression of ACE2 and furin in colon carcinomas. ACE2 staining is located in the cytoplasm and cell membrane; the intensity is very strong and observed in all colon cancer cells, (**A**,**B**). Furin staining is located in the cytoplasm in a granular pattern; the intensity is moderate to strong and is observed in all colon cancer cells, (**C**,**D**).

**Figure 3 cancers-14-02582-f003:**
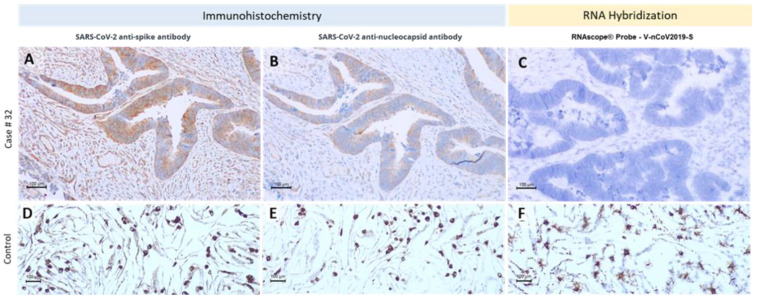
SARS-CoV-2 detection on a colon carcinoma case. Patient no. 32 tested positive for SARS-CoV-2 after sigmoidectomy surgery and we observed staining in the cancer cell with both SARS-CoV-2 antibodies (**A**,**B**) but not with in situ hybridization (ISH) (**C**). We extracted the RNA from the tumor block; the PCR results were positive. The lower panel shows infected VERO cells used as the positive control for antibodies (**D**,**E**) and ISH (**F**).

**Figure 4 cancers-14-02582-f004:**
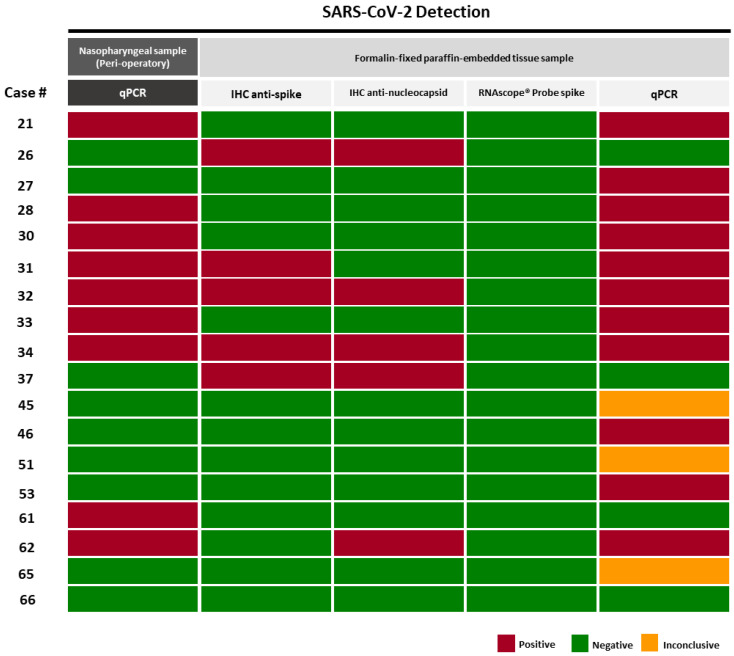
Comparison between SARS-CoV-2 detection by immunohistochemistry, in situ hybridization, and RT-qPCR in FFPE tumor tissues. Red—Positive cases; Green—Negative cases; Yellow—Inconclusive cases.

**Table 1 cancers-14-02582-t001:** Association between the expressions of ACE2 and furin in colon, gastric, and thyroid carcinomas.

			ACE2 Expression		
			Positive (*n*)	Negative (*n*)	*p* Value
Furin expression	Colon carcinoma	Positive	49	6	0.128
		Negative	3	2	
	Gastric carcinoma	Positive	13	14	0.552
		Negative	1	2	

## Data Availability

The datasets generated during and/or analyzed during the current study are available from the corresponding author on reasonable request.

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
