# Peer review of "Searching for SARS-CoV-2 in Cancer Tissues: Results of an Extensive Methodologic Approach based on ACE2 and Furin Expression"

_cancers, 2022, doi:10.3390/cancers14112582_

Round 1

Reviewer 1 Report

The aim of the paper “ACE2 and Furin Expression Does Not Predict SARS-CoV-2 Infection in the Cancer Tissue: Results of an Extensive Methodologic Approach” is the investigation of the expression of ACE2 and furin in tumour tissues and adjacent areas, some of them belonging to patients who tested positive for SARS-CoV-2, using a complex methodology, involving cell line infection, electron microscopy, immunohistochemistry, RNA in situ hybridization, RNA extraction and RT-qPCR, and statistical analysis.

Broad comments:

The manuscript has an original conceptualization, is well organized, adding value to the previous research concerning SARS-CoV-2 infection. The references comprise 29 recent articles from the last two pandemic years. The findings are illustrated in three tables and seven figures.

Specific comments:

The sections and subsections of the manuscript are appropriate.

In 1. Introduction section, the authors have presented information regarding COVID-19 epidemiology, pathogenesis and involvement of ACE2 and furin, aim of the study, specific goals and study design.

However, the authors have added the results of the study in this section. It is recommended to delete the results of the study from Introduction section and to introduce them in their dedicated section (3. Results) or to reformulate them with the authors expectations regarding the possible results of their study.

In 2. Material and Methods, the authors have described all the process of patients’ selection, cell line infection, electron microscopy, immunohistochemistry, RNA in situ hybridization, RNA extraction and RT-qPCR, and statistical analysis.

However, there is no mention regarding the compliance to ethics principles.

Moreover, the utility of cell line infection should be clearly explained.

In 3. Results section, the authors have described their findings in two subsections (3.1 ACE2 and Furin are highly expressed in Colon and Gastric carcinomas and 3.2 Different methodologic approaches to detect SARS-CoV-2 in tumor samples), using 3 tables and seven figures.

However, some of the figures and tables are not count as 1, 2..., but as S1, S2,…, without any explanatory reason. It is recommended to use the same system of numbering for all tables and figures.

Moreover, some figures lack subdivision in A and B (although there are some special references in the text of the manuscript). It is recommended to add A and B in the caption of Figure 1, in images and caption of Figure 2 and S1-S4.

Furthermore, "ACE" should be replaced by "ACE2" in the text of page 6, "n.a." should be explained in Table 1, and “Thyroide” should be replaced by “Thyroid” in graphic representation of Figure S1.

Moreover, the presentation of the clinicopathological features of the cases, presented as “data not shown” would be necessary, at least the mention of which characteristics were taken into consideration and led to inconclusive correlations with ACE2 and furin expression.

In 4. Discussion section the findings are compared with other literature information. However, the work of other authors is variable introduced, such as “Hikmet et al”, “Baldelli R et al”, “Narayan S. et al, “, and these should be corrected.

A strong characteristic is that there are important references to the limitations of the study regarding the methodology and patients’ selection, which may be responsible for the failure in demonstration of SARS-CoV-2 in tumour samples.  

Furthermore, the section 5. Conclusions highlights the confirmation of their hypothesis versus limitations given by the relatively small size of the series of cases and sets up future directions of study in this domain, by “more reliable IHC or ISH studies”.

Author Response

Dear Editor,

We are submitting a revised version of the Manuscript (ID: cancers-1697047) entitle “ACE2 and Furin Expression Does Not Predict SARS-CoV-2 Infection in the Cancer Tissue: Results of an Extensive Methodologic Approach”. We have addressed all the questions raised by the reviewers as thoroughly as possible. The alterations in the manuscript were performed and the changes in the main manuscript and supplementary data document are highlighted in yellow color.

Reviewer 1 recognizes the value of this manuscript to the previous research on SARS-CoV-2 infection and suggests some alterations that we answer point-by-point:

  1. We eliminated the results from the introduction section as suggested.
  2. We added a paragraph regarding the compliance to ethical principles in the Methods section. We introduced the following sentence in the Materials and Methods section (Point 2.1. Patient selection): “This study was conducted in accordance with the Declaration of Helsinki. The protocol was approved by the Ethics Committee of Hospital das Clínicas da Faculdade de Medicina da Universidade de São Paulo, Brazil (Project identification code: 4.338.488). The use of retrospective material from human samples for research purposes in Portugal is legally foreseen in the Portuguese law (Decreto-Lei n. º 274/99).”
  3. Since SARS-CoV-2 is a new virus and the methodologies to identify it are also novel, we decided to introduce an infected cell line, with high viral loads, that was collected from cell culture and then formalin-fixed paraffin-embedded to have the same pre-analytic conditions as tumor tissues retrieved from pathology archives. These infected cells and placenta tissue were paramount as positive controls to a variety of technical approaches that were used in this manuscript.
  4. The table and figure numeration were identified as main and supplementary material. Tables and figures with an “S” before the number correspond to supplementary material. We introduced a sequence of letters in the figures and changed the legends accordingly (Figures 2 and 3, as well as figure legends, were edited). We corrected the errors identified by the reviewer (ACE2, Thyroid, and n.a. specification)
  5. Regarding the clinical and pathological data, we evaluated the correlation of ACE2 and Furin expression with tumor size, lymph node metastasis, distant metastasis, residual tumor, invasion, and presence of inflammatory infiltrate but we did not find any statistically significant correlation. We described these clinicopathological features in the revised ms text (Page 7, line 261).
  6. We corrected the literature information in the discussion

Reviewer 2 Report

Dear authors:

The authors submitted an original article, entitled " ACE2 and Furin Expression Does Not Predict SARS-CoV-2 Infection in the Cancer Tissue: Results of an Extensive Methodologic Approach" to Cancers for our further peer review.   

Generally, the authors tested expression of ACE2 and furin in tumor samples (FFPE) of 90 patients with colonic (n=60), gastric (n=19) and thyroid (n=11) carcinomas via results of immunohistochemical expression, in situ hybridization, and RT-qPCR to investigate if SARS-CoV-2 is able to infect cancer cells, and if this is associated with the expression of ACE2 and furin.

            The topic is interesting, although the protein expression of ACE2 and furin had been widely discussed in a variety of articles. Of importance, the authors used clinical tumor samples and non-tumor parts with comparison of immunohistochemical expression, in situ hybridization, and RT-qPCR. The study can be improved. Moreover, major concerns are listed as follows.

  1. The 4th paragraph of the introduction, the authors mentioned, “based on previous data showing that tumor cells can express this receptor for the virus” without any citation. Please cite your article precisely. Besides, the header of word document was wrong, since it showed “Cancers 2021, 13, x FOR PEER REVIEW”.

  2. In the 2.1 patient selection, the approval for Institutional Review Board should be provided, since this is a study with human tissues.

  1. In the 2.4 immunohistochemistry, the evaluation and interpretation criteria for different proteins should be mentioned in detail, including their intensity and density.

  1. In the 2.7 statistical analysis, the analytical strategies should be well explained. The Pearson chi-square test is usually not recommended for testing the composite hypothesis of normality due to its inferior power properties compared to other tests. It is a common practice to compute the p-value from the observed data to a model that distributes the data according to the expectation that the variables are independent. The protein expression of ACE2 and furin should be proven independent, so that the authors can use Pearson chi-square test. Moreover, in the Table S1, the p value was most likely from the fisher’s exact test. It should be explained. In addition, the authors didn’t explain the significance of p value and one-tailed/two-tailed problems.

  1. In Figure S1, the authors provided two pictures of furin IHC stains in non-neoplastic gastric tissue, but they didn’t provide any normal thyroid pictures. Please provide all representative immunostains in three non-neoplastic tissues.

  1. In Figure S2, the figure legend should be re-written and the figure labeling of Vero cells and human placenta tissue should be clearly illustrated.

  1. In the abstract, the authors mentioned,” from the subseries of 18 cases, 11 tested positive by PCR detection.” But, in the second paragraph of session 3.2, the authors said, “The subset of 18 cases we selected were 9 positive and 9 negative for SARS-CoV-2, by nasopharyngeal PCR test in the peri-operatory period.” The statement was totally different, and please correct the problem.

  1. If the “ACE2 receptor” in Figure 1A and “ACE2” in Figure 2 are the same antibody, please label the antibody consistently. Moreover, the description in the session 3.1 showed “ACE expression was observed in the cytoplasm and cell membrane of tumor cells and furin staining was limited to the cytoplasm in a granular pattern (Figure 2).” It obviously omitted “2” and made a “ACE”.

  1. In session 3.1 and Table 1, the association between expression of ACE2 and furin failed to show statistical significance. It is probably associated your limited cancer samples, since the trend of simultaneous expression in colon and stomach is remarkable. An increase of colonic and gastric tumor samples should be considered, since this inconclusive result contributed nothing to scientific value.

  1. In Figure S4, the spike SARS-CoV-2 proteins had prominent nonspecific staining in smooth muscles, lymphoplasma cells in lamina propria, and endothelial cells, while the nucleocapsid SARS-CoV-2 proteins showed nonspecific staining in lymphoplasma cells. These nonspecific staining may contribute largely to our interpretation, and please explain how to avoid or lower the background staining in the discussion.

  1. In session 3.2, only 18 cases had results of nasopharyngeal PCR test in the peri-operatory period. In situ hybridization was negative in all cases. The case limitation could be the major cause for a direct association between ACE2 expression and SARS-CoV-2 infection, although anti-spike and anti-nucleocapsid antibodies revealed a concordance expression. If the authors didn’t want to increase cases, please consider revise your topic, “ACE2 and Furin Expression Does Not Predict SARS-CoV-2 Infection in the Cancer Tissue.” Since this failure of prediction may be resulted from small case number and the lack of peri-operative nasopharyngeal PCR test, the conclusive title may be inappropriate.

Author Response

Dear Editor,

We are submitting a revised version of the Manuscript (ID: cancers-1697047) entitle “ACE2 and Furin Expression Does Not Predict SARS-CoV-2 Infection in the Cancer Tissue: Results of an Extensive Methodologic Approach”. We have addressed all the questions raised by the reviewers as thoroughly as possible. The alterations in the manuscript were performed and the changes in the main manuscript and supplementary data document are highlighted in yellow color.

Reviewer 2 recognizes the interest of the topic but makes suggestions for improvement. We answer each of the reviewer’s concerns point-by-point:

  1. We introduced the citation (Reference 14) in the first sentence of the 4th paragraph of the introduction

  1. We added a paragraph regarding the compliance to ethical principles in the Methods section as also indicated by Reviewer 1. We introduced the following sentence in the Materials and Methods section (Point 2.1. Patient selection): “This study was conducted in accordance with the Declaration of Helsinki. The protocol was approved by the Ethics Committee of Hospital das Clínicas da Faculdade de Medicina da Universidade de São Paulo, Brazil (Project identification code: 4.338.488). The use of retrospective material from human samples for research purposes in Portugal is legally foreseen in the Portuguese law (Decreto-Lei n. º 274/99).”

  1. We introduced the following information regarding the scoring system in section 2.4: “Immunocytochemistry results were evaluated by, at least, two independent observers (SR, FS, DM and/or PC) that register the staining intensity [(0) no staining; (1) week staining; (2) moderate staining; and, (3) strong staining)] and the percentage of cells stained [(0)0%; (1)-0-1%; (2) 2-10%; (3) 11-33%; (4) 34-66%; (5) 67-100%]. Then, a score was calculated by summing the intensity and extension values. We considered nega-tive when the score was <2 and positive when the score was >3.”

  1. The Reviewer suggests a more detailed explanation of the statistical analysis performed in the ms. After a more careful analysis of the data and a biostatistical consultation, we decided in accordance with the reviewers, and we used Fisher’s exact test to compute the p-value as we have observations <5 in our correlation tables. We made changes in the tables and in the Material and methods section accordingly.

  1. We added normal thyroid immunostaining with ACE2 and Furin in Supplementary Figure S1.

  1. We have re-written the legend of the supplementary figure, introduced letters to each photo, and clarified the immunostaining and ISH performed in VERO cells and placenta tissue.

  1. In the abstract, the Reviewer comments that the authors mentioned,” from the subseries of 18 cases, 11 tested positive by PCR detection.” But, in the second paragraph of session 3.2, the authors said, “The subset of 18 cases we selected were 9 positive and 9 negative for SARS-CoV-2, by nasopharyngeal PCR test in the peri-operatory period.” The statement was totally different, and please correct the problem. The first sentence refers to the PCR performed in tumor samples (11 tested positive). In 3.2 section we describe the method of selection of these 18 cases (we selected 9 PCR(+) and 9 PCR(-) patients in which we had the result performed in nasopharyngeal samples). We added this information in the abstract to clarify this issue: “From the subseries of 18 cases, 11 tested positive by PCR detection performed in tumor blocks”

  1. Figure 2 was edited. “ACE2 receptor” was replaced by ACE2. Figure 2 legend was also corrected.

  1. In session 3.1 and Table 1, the association between expression of ACE2 and furin failed to show statistical significance. Reviewer suggests this is probably associated with the limited cancer samples since the trend of simultaneous expression in colon and stomach is remarkable. And also suggests that increasing the colonic and gastric tumor samples should be considered and would contribute to the scientific value of the Ms. We totally agree with this comment. However, these cases were selected in a time frame where the surgical procedures were performed regardless of the patient’s SARS-CoV-2 infection state and therefore we cannot replicate these conditions to enlarge the number of samples. However, it is of relevance for future analysis where enlarged series should address the same questions as suggested in the current Ms.

  1. In Figure S4, the spike SARS-CoV-2 proteins had prominent nonspecific staining in smooth muscle, lympho-plasma cells in the lamina propria, and endothelial cells, while the nucleocapsid SARS-CoV-2 proteins showed nonspecific staining in lympho-plasma cells. The Reviewer suggests that this nonspecific staining may contribute to our interpretation, and asks for an explanation on how to avoid or lower the background staining. We acknowledge that the production of specific antibodies is essential to identify with precision the antigen epitope of interest. At the time (and still at present), the options for primary antibody selection were limited and tested in a few laboratories necessarily with different working conditions. The main pillar to identify viral staining with specificity in our study was the use of infected VERO cells, which gave us a gold standard, included in every slide for the recognition of virus specificity. The nonspecific staining is unfortunately hardly resolved in the current lab setting. We addressed this issue in the discussion of the revised ms “The production of specific antibodies is essential to identify with precision the protein epitope of interest. In fact, IHC detection of SARS-CoV-2 is still under scrutiny and antibody specificity is limited with immunostaining being reported in negative control tissues [28]. The main pillar for staining specificity in our current study was the use of infected VERO cells that gave us a gold standard, included in every slide, for the specificity of viral recognition.”

  1. Reviewer 2 suggests a revision on the Ms title “ACE2 and Furin Expression Does Not Predict SARS-CoV-2 Infection in the Cancer Tissue”, since a conclusive title may be inappropriate. In fact, the failure of prediction may result from the small number of cases and the lack of peri-operative nasopharyngeal PCR tests. We accordingly modified the title to: “Searching for SARS-CoV-2 in Cancer Tissues: Results of an Extensive Methodologic Approach based on ACE2 and Furin Expression”

Round 2

Reviewer 2 Report

The authors kindly submitted submitted a revised version of the original article entitled, “Searching for SARS-CoV-2 in Cancer Tissues: Results of an Extensive Methodologic Approach based on ACE2 and Furin Expression.” In general, the authors retrospectively tested expression of ACE2 and furin in tumor samples (FFPE) of 90 patients with colonic (n=60), gastric (n=19) and thyroid (n=11) carcinomas via results of immunohistochemical expression, in situ hybridization, and RT-qPCR to investigate if SARS-CoV-2 is able to infect cancer cells, and if this is associated with the expression of ACE2 and furin. Although the novelty is not high, the case number is small, and there are published articles focusing on similar issues, they provided insights of clinical tumor samples and non-tumor parts. The manuscript has been revised according to reviewers’ comments. However, there are minor comments that they need to revise.

  1. In the 2.7. Statistical analysis, P value <.05 was considered statistically significant. But the authors didn’t mention it as one-tailed or two-tailed.

  1. In the line 385 of the discussion, there is a typo in the following sentence, “The most plausible interpretation for negative results in all our PFPE carcinoma cases is that ISH…..”
  2. In Table 2, among the 9 cases with nasopharyngeal PCR-negativity before surgery, 3 (33.3%) were tested positive in qPCR for tumor tissues and 3 (33.3%) were inconclusive. Could the authors explain and put into discussion why the qPCR for tumor samples had a higher positivity than nasopharyngeal qPCR? Since the tumor samples were obtained from the surgery, it might indicate a prior infection or process contamination. Please explain the finding in detail and in the discussion.

Author Response

May 17th 2021

Dear Editor,

We are submitting a revised version of the Manuscript (ID: cancers-1697047) at first with the title “ACE2 and Furin Expression Does Not Predict SARS-CoV-2 Infection in the Cancer Tissue: Results of an Extensive Methodologic Approach” but now re-submitted with the title “Searching for SARS-CoV-2 in Cancer Tissues: Results of an Extensive Methodologic Approach based on ACE2 and Furin Expression”

Reviewer 2 suggests the correction of some minor corrections that we are sending back in a revised version highlighted in green:

  1. We clarified the statistical analysis performed and the p-value that was obtained from a two-tailed testing. (Page 5, line 236)
  2. We corrected the typo in line 385, page 10.
  3. We introduced a full paragraph, discussing the discrepancies obtained with qPCR technique in nasopharyngeal and FFPE tumor samples (Page 10, line 382-391): “The six cases that tested positive or inconclusive in qPCR from the surgical speci-men despite a negative nasopharyngeal test should cautiously be considered as real discrepancies since the nasopharyngeal test was performed two days before surgery and therefore intra-hospital infection can not be discarded. Also, the negative naso-pharyngeal test does not exclude the presence of a SARS-CoV-2 infection because the sample collection at nasopharynx could have been performed during an early stage of infection, translating into low viral loads and subsequently in undetectable RNA levels. In addition, qPCR tests at hospital and on tumor samples were performed on di-ferent materials whose relative quantities are difficult to estimate and using qPCR kits with different efficiency/sensitivity.”

Thank you

Fernando Schmitt